# Nonsurgical Periodontal Treatment Options and Their Impact on Subgingival Microbiota

**DOI:** 10.3390/jcm11051187

**Published:** 2022-02-23

**Authors:** Susanne Schulz, Jamal M. Stein, Anne Schumacher, David Kupietz, Sareh S. Yekta-Michael, Florian Schittenhelm, Georg Conrads, Hans-Günter Schaller, Stefan Reichert

**Affiliations:** 1Department of Operative Dentistry and Periodontology, Martin-Luther-University Halle-Wittenberg, 06108 Halle, Germany; annepeuthert@live.de (A.S.); dkupietz@arcor.de (D.K.); hans-guenter.schaller@uk-halle.de (H.-G.S.); stefan.reichert@uk-halle.de (S.R.); 2Department of Operative Dentistry, Periodontology and Preventive Dentistry, University Hospital (RWTH) Aachen, 52074 Aachen, Germany; jstein@ukaachen.de (J.M.S.); samichael@ukaachen.de (S.S.Y.-M.); f.schittenhelm@gmx.de (F.S.); gconrads@ukaachen.de (G.C.); 3Private Practice, 52062 Aachen, Germany

**Keywords:** periodontitis stage III/IV, nonsurgical periodontal treatment, quadrant-wise debridement, full-mouth scaling, full-mouth disinfection, full-mouth disinfection with adjuvant erythritol air-polishing, subgingival microbiota, *Porphyromonas gingivalis*, *Eubacterium nodatum*, *Prevotella dentalis*

## Abstract

Background: Different periodontal treatment methods (quadrant-wise debridement, scaling and root planing (Q-SRP), full-mouth scaling (FMS), full-mouth disinfection (FMD), and FMD with adjuvant erythritol air-polishing (FMDAP)) were applied in periodontitis patients (stage III/IV). The study objective (substudy of ClinicalTrials.gov Identifier: NCT03509233) was to compare the impact of treatments on subgingival colonization. Methods: Forty patients were randomized to the treatment groups. Periodontal parameters and subgingival colonization were evaluated at baseline and 3 and 6 months after treatment. Results: Positive changes in clinical parameters were recorded in every treatment group during the 3-month follow-up period, but did not always continue. In three groups, specific bacteria decreased after 3 months; however, this was associated with a renewed increase after 6 months (FMS: *Porphyromonas gingivalis*; FMD: *Eubacterium nodatum*, *Prevotella dentalis*; and FMDAP: uncultured *Prevotella* sp.). Conclusions: The benefit of all clinical treatments measured after 3 months was associated with a decrease in pathogenic bacteria in the FMS, FMD, and FMDAP groups. However, after 6 months, we observed further improvement or some stagnation in clinical outcomes accompanied by deterioration of the microbiological profile. Investigating the subgingival microbiota might help appraise successful periodontal treatment and implement individualized therapy.

## 1. Introduction

Periodontitis, a prevalent inflammatory disease in humans, develops as a consequence of the interplay between subgingival dysbiosis of the oral microbiota and the resulting destructive host-immune response [1,2]. Among the possible therapy options, mechanical debridement applying scaling and root planing (SRP) represents an integral component of successful periodontal therapy [3]. To increase the effectiveness of classical quadrant-wise SRP (Q-SRP), various complementary therapy options have been discussed and evaluated.

It is known that pathobionts can colonize almost all niches of the oral cavity; therefore, a quadrant-wise approach is associated with the risk of recolonization [4]. To minimize this risk, full-mouth scaling (FMS) comprising SRP of all quadrants within 24 h was introduced. Supporting this, a full-mouth disinfection (FMD) approach has been proposed (combination of FMS and complementary chlorhexidine treatment) [5,6,7]. However, the evidence showing a clinical benefit of additional therapy options is inconsistent [8,9,10]. In a Cochrane Review, the authors conclude that there is no significant clinical benefit for FMS and FMD compared to Q-SRP [11]. Furthermore, the positive influence of additional mechanical intervention on the biofilm for clinical outcome was discussed. In this context, it was suggested that applying erythritol air-polishing (FMDAP) in addition to nonsurgical periodontal therapy might improve both clinical and microbiologic outcomes [12]. 

In the original study on which this substudy is based, Stein and colleagues demonstrated that FMDAP indeed enhances clinical outcome compared to SRP [13]. 

In addition, the importance of rehabilitating the subgingival dysbiosis has long been considered another indicator of successful nonsurgical periodontal therapy [14]. However, most studies focused only on select bacteria associated with periodontitis, with heterogeneous results (*Porphyromonas gingivalis* (*Pg*), *Prevotella intermedia*, *Aggregatibacter actinomycetemcomitans* (*Aa*) [14]; *Aa*, *Pg*, *Tannerella forsythia* (*Tf*), *Treponema denticola* (*Td*), *Dialister pneumosintes* [15]; *Pg*, *Tf*, *Td* [16]; and *Pg*, *Tf*, *Fusobacterium nucleatum* [17]).

To date, only few studies have taken a broader approach and investigated the impact of nonsurgical periodontal therapy on multiple subgingival pathogens. Rosalem et al. applied checkerboard DNA-DNA hybridization (40 subgingival species) [18]. They identified a reduction in bacteria of the orange and red complexes and an increase in Actinomyces species after periodontal therapy. In initial studies applying a comprehensive approach, next-generation sequencing (NGS) was used to investigate complex subgingival microbial changes due to nonsurgical periodontal therapy but without comparing different therapies [19,20,21].

The present study was initiated to narrow this gap. Consequently, the aim of this work was to compare the microbiological outcome of different, clinically successfully applied nonsurgical periodontal therapies (Q-SRP, FMD, FMS, and FMDAP) in patients with periodontitis stage III/IV using NGS methods. One hypothesis of the study was to expect no considerable differences in microbiological outcome comparing the four treatment options. Our second assumption, however, was that a considerable shift in biofilm composition would develop along with clinical improvement in the periodontium after mechanical treatment with all four treatment options. 

## 2. Materials and Methods

### 2.1. Study Design

The present work comprises a substudy of a registered clinical trial database study “Anti-infectious Therapy of Periodontitis—Comparison of Different Clinical Strategies” (ClinicalTrials.gov: NCT03509233). The study was approved by the ethics committee of the University of Aachen, Germany (EK 046/16). Informed written consent was obtained from each patient. All investigations were carried out in accordance with the ethical guidelines of the “Declaration of Helsinki” and its amendment in “Tokyo and Venice”. 

The study was designed as a prospective, randomized, blinded, four-arm, parallel-group study [11]. Patients were enrolled at two university dental departments (Department of Operative Dentistry and Periodontology, Martin-Luther-University, Halle/S, and Department of Operative Dentistry, Periodontology and Preventive Dentistry, University Hospital (RWTH), Aachen, Germany) after periodontal assessment and radiological examination. The present study included 40 patients, 10 in each treatment group. 

### 2.2. Patients

Patient characteristics and treatment strategies (including calibration of the examiners) were described in detail by Stein et al., 2021 [13]. Patients were enrolled in the study between March/2017 and May/2020. 

For inclusion in the study, patients had to be suffering from periodontitis stage III/IV [22]. They had to have at least 18 teeth, two multi-rooted and/or two single-rooted teeth in the first quadrant, with at least six sites with a probing depth (PPD) ≥ 6 mm. Radiological assessment had to show bone loss extending to the middle or apical third of the root. 

Exclusion criteria were pregnancy, inability to give written informed consent, subgingival scaling and root planing within the last 6 months, use of antimicrobial rinsing solutions or intake of systemic antibiotics during the last 4 months, and intake of drugs that potentially cause gingival hyperplasia (e.g., hydantoin, nifedipine, and cyclosporin A).

Patients were classified as smokers if they smoked more than 10 cigarettes/day for more than 5 years [23,24]. All patients underwent supragingival debridement as an initial treatment. Patients received oral hygiene instructions including interdental care (at baseline and 3 and 6 months). Patients were given the same toothpaste and toothbrushes (meridol^®^; CP-GABA, Hamburg, Germany) and interdental brushes (elmex^®^, CP-GABA, Hamburg, Germany). 

In the next step, the patients were randomized into four treatment groups [13]. For this purpose, sealed envelopes containing a code with the group allocation were used. An independent assistant had already randomized the patients before the appointment was made. In order to randomize for smoking, two different sets were used for each treatment group, an independent randomization assistant randomly drawing an envelope from the two sets for smokers and nonsmokers. The randomization assistant gave the allocation details only to the dentist who performed the treatment; this information was not reported to the clinical investigator. Each of the four treatment groups included the first 10 patients who successfully completed all follow-up examinations (3 months, 6 months) at the study centers in Halle and Aachen. The treatment groups were comprised of Q-SRP (quadrant-wise subgingival scaling and root planing clockwise in four sessions with an interval of 1 week between each quadrant without the use of antiseptics), FMS (full-mouth SRP within 24 h without the use of antiseptics), FMD (full-mouth SRP within 24 h with disinfection using antiseptic applications according to the protocol of Quirynen et al. (1998) [6]), and FMDAP (FMD combined with the use of subgingival erythritol air-polishing).

### 2.3. Clinical Periodontal Parameters

The dental examination involved determining the plaque index (PI) and gingival index (GI) [25], probing depth (PPD), clinical attachment level (CAL), and bleeding on probing (BOP) at six sites around each tooth (Parodontometer DB764R UNC 15; Aesculap, Tuttlingen, Germany). In addition, the proportion of nonbleeding, shallow sites (PPD ≤ 4 mm without bleeding on probing) according to the current periodontitis classification [22], was determined. All periodontal examinations were performed by one blinded examiner at each center at baseline and at 3 and 6 months after therapy. The comprehensive study protocol is available from the authors.

### 2.4. Statistical Analysis

Statistical analyses were performed using SPSS software (SPSS v.25.0 package; IBM, Chicago, IL, USA). Continuous data were evaluated for normal distribution using the Kolmogorov–Smirnov test and the Shapiro–Wilk test. Values of *p* < 0.05 were considered statistically significant. Differences in periodontal indices within the treatment groups compared to baseline were evaluated using the *t*-test (paired samples; Tukey–Kramer correction). The significance of differences in dental parameters among treatment groups at each time point was assessed by applying analysis of variance (ANOVA).

Differences in alpha diversity (Shannon indices) and beta diversity (UniFrac analyses) were assessed by Wilcoxon signed-ranks test and corrected for multiple comparisons using the Benjamini–Hochberg false discovery rate (FDR) of 5%. Differences in means among treatment groups (at all phylogenetic levels) were evaluated using a paired *t*-test, including FDR analysis.

### 2.5. Microbial Subgingival Sample Collection and DNA Isolation

Microbial samples were obtained from the deepest pocket of each quadrant at baseline (before periodontal therapy was performed) and after 3 and 6 months. All four bacterial plaque samples were pooled and microbial DNA was extracted using QIAamp^®^ DNA mini kit (Qiagen, Hilden, Germany). 

After verifying the quality and quantity (Qubit^®^ dsDNA BR Assay Kit (ThermoFisher Scientific, Schwerte, Germany) and Qubit^®^ Fluorometer (ThermoFisher Scientific, Schwerte, Germany); Bioanalyzer DNA 1000 chip (Agilent, Santa Clara, CA, USA), the DNA was shipped to Novogene Co., Ltd., Cambridge, UK for further analysis on dry ice.

### 2.6. Next Generation Sequencing

For NGS (Novogene Co., Ltd.), 16S V3/V4 regions were targeted to generate amplicons. Phusion^®^ High-Fidelity PCR Master Mix (New England Biolabs, Ipswich, MA, USA) was used for amplification. PCR products were purified using the Qiagen Gel Extraction Kit (Qiagen, Hilden, Germany). Library preparation and indexing were carried out using NEBNext^®^ UltraTM DNA Library Pre Kit (Illumina, San Diego, CA, USA). To quantify and normalize the library, the DNA was evaluated using a Qubit^®^ 2.0 Fluorometer (Thermo Scientific, Waltham, MA, USA) and Agilent Bioanalyzer 2100 system (Agilent, Santa Clara, CA, USA). For sequencing analysis, the Illumina platform (Illumina, San Diego, CA, USA) was applied and 250 bp paired-end reads were generated.

### 2.7. Data Analysis

#### 2.7.1. Paired-End Read Assembly

The reads were merged using FLASH (V1.2.7, http://ccb.jhu.edu/software/FLASH (accessed on 7 May 2021)) [26]. The raw tags were preprocessed and the quality filtered according to the QIIME workflow (V1.7.0, http://qiime.org/index.html (accessed on 7 May 2021)) [27]. Using the UCHIME algorithm, the tags were compared with a reference database for chimera depletion (http://www.drive5.com/usearch/manual/uchime_algo.html (accessed on 7 May 2021); http://drive5.com/uchime/uchime_download.html (accessed on 7 May 2021)) [28]. 

#### 2.7.2. Operational Taxonomic Unit (OTU) Cluster and Species Annotation

Uparse software (Uparse V7.0.1001, http://drive 5.com/uparse (accessed on 7 May 2021)) was applied for sequence analysis [29]. Sequences showing >97% similarity were assigned to the same OTU.

For species annotation, we referred to the GreenGene Database (http://greengenes.lbl.gov{cgi-bin{nph-index.cgi (accessed on 7 May 2021)) based on the RDP classifier algorithm (V2.2, http://sourceforge.net/projects/rdp-classifier (accessed on 7 May 2021)) [30]. OTU abundance information was normalized according to a standard sequence corresponding to the sample with the least sequences. All subsequent analyses were assessed using output-normalized data. 

#### 2.7.3. Bioinformatics: Alpha and Beta Diversity

Alpha diversity was calculated with QIIME (V1.7.0) and displayed with R software (V2.15.3). The Shannon index was used to identify community diversity (http://mothur.org.wiki/Shannon (accessed on 7 May 2021)). 

Beta diversity on both weighted and unweighted UniFrac was also calculated by applying the QIIME software (V1.7.0). The general distribution of the resulting bacterial community composition was evaluated using principal coordinates analysis (PCoA) with the R package ggplots2 (V2.15.3). The linear discriminant analysis effect size (LEfSe) pipeline was applied according to Galaxy (https://huttenhower.sph.harvard.edu/galaxy/ (accessed on 11 May 2021), https://huttenhower.sph.harvard.edu/galaxy/datasets/7c8874d23329d3dd/display/?preview=True (accessed on 11 May 2021)).

## 3. Results

### 3.1. Patient Characteristics

Patients were recruited from March/2017 until May/2020. The first 10 patients in each treatment group (40 patients in total) who completed the 3- and 6-month follow-up in Halle and Aachen were included in this substudy. Demographic parameters at baseline are displayed for all treatment groups in Table 1 (Table 1). Table 2 compares the full-mouth clinical parameters at baseline as well as at 3 and 6 months after periodontal treatment. In general, an improvement in periodontal parameters was documented after each of the four periodontal regimens (Table 2). A comparison of the different treatment groups showed that the positive changes were comparable in each group regarding reduction in PI, PPD, GI, and clinical attachment gain. However, there was no consistent improvement in all clinical parameters after 6 months (Table 2). 

### 3.2. Microbiota Structure Analysis

The periodontal microbial composition was evaluated by comparing subgingival samples of periodontal patients randomized to one of four treatment procedures (Q-SRP, FMS, FMD, or FMDAP).

#### 3.2.1. Relative Abundance in Relation to Periodontal Therapy

According to the species annotation results, the distribution within the different treatment groups was determined at the phylogenetic level over time. In Figure 1 the results are displayed in bar graphs for the top ten most abundant genera with (Figure 1). Among them, *Fusobacterium*, *Prevotella*, *Porphyromonas*, *Treponema*, and *Streptococcus* were found in the top five positions. Investigating the time course after periodontal therapy, no statistically significant differences were identified among these genera in all the periodontal treatment groups. However, a decrease in *Fusobacterium* (all treatment groups) and *Treponema* (except FMDAP group) was initially recorded 3 months after therapy, but this was associated with an increase after 6 months for *Fusobacterium* (Q-SRP, FMD, and FMDAP) and *Treponema* (FMD). However, the 6-month values remained slightly improved compared to the baseline values.

Interestingly, the pattern of *Streptococcus* was almost the opposite of that described for *Fusobacterium*.

#### 3.2.2. Alpha Diversity of Subgingival Microbiota Related to Periodontal Treatment

Differences in the alpha diversity within treatment groups were evaluated using the Shannon index (Figure 2). In conclusion, alpha diversity was not significantly changed by periodontal treatment after 3 and after 6 months. However, trends were noted. The alpha diversity in the subgingival microbiota of patients undergoing Q-SRP and FMDAP procedures decreased continuously. However, a comparable trend could not be observed for the other treatment groups. While alpha diversity initially increased in the FMS group, it decreased again after 6 months to almost the baseline level. In the FMD group, a continuous increase over time was recorded. 

#### 3.2.3. Effects of Periodontal Treatment on Beta Diversity

UniFrac is a widely used instrument to evaluate beta diversity, which involves the assessment of differences between communities. When considering the abundance of each taxon in a community, quantitative measures such as weighted UniFrac can be applied. When comparing community differences it was obvious that UniFrac distances increased in the FMS and FMDAP group over time after periodontal therapy (Figure 3). In the Q-SRP group a similar trend was shown. Only in the FMD treatment group was an initial increase in UniFrac distances observed 3 months after therapy combined with a decrease after another 3 months. 

In the Q-SRP group, almost no significant deviation was shown at any of the phylogenetic levels. In the FMS group, changes were observed at all phylogenetic levels. At the species level, *Porhyromonas gingivalis* was significantly less prevalent after 3 months than at the time of treatment (*p* = 0.013). However, 6 months after treatment, the occurrence of this bacterium had increased again but was below the initial level (n.s.). In contrast, the occurrence of *Campylobacter concisus* had increased steadily 3 months (*p* = 0.040) and 6 months (*p* = 0.021) after treatment. 

Patients undergoing FMD treatment initially showed a significant decrease in the occurrence of *Eubacterium nodatum* (*p* = 0.038) and *Prevotella dentalis* (*p* = 0.015) 3 months after treatment. However, colonization of these bacteria increased again 6 months after treatment, but did not reach the initial level (n.s.) (Table 3).

In the FMDAP group, the uncultured *Prevotella* sp. showed a course very similar to that of the two aforementioned bacteria in the FMD group; initially, a decrease (n.s.) was combined with a renewed increase (*p* = 0.025) (Table 3).

In PCoA, no discrimination of bacterial communities within the treatment groups was obvious regarding the different time points.

To illustrate the distribution of relative abundances within the treatment groups at different points in time, linear discriminant analysis (LDA) and LEfSe were applied. Here, we used these models for ranking bacterial species based on relative differences within the treatment groups. However, based on an LDA score of <4.0, no differences in the presence of bacteria at the phylogenetic species level were shown within any of the periodontal treatment groups. 

## 4. Discussion

An important goal that has already been addressed in clinical trials is the issue of successful nonsurgical treatment of periodontitis [11,31]. In addition to improving the clinical periodontal parameters, the treatment effect on a shift in the subgingival bacterial spectrum is of great interest. In the present study, four different nonsurgical periodontal treatment strategies (Q-SRP, FMS, FMD, and FMDAP) were evaluated regarding changes in microbiological composition. This study is, to our knowledge, the first longitudinal study comparing the subgingival microbial outcome of different treatment options over a 6-month period. A profound change in subgingival microbial composition 3 months after treatment (FMS, FMD, and FMDAP) was shown. However, 6 months after treatment, a trend towards a profile that includes fewer disease-associated bacteria was again noticeable. Nevertheless, the microbial outcomes remained better than the baseline results. 

In most studies investigating the microbiological effect of nonsurgical periodontal treatment, the focus has been on individual pathogens [14,15,16,17]. Newer analytical methods, such as 16S rRNA gene sequencing, can assess ecological changes that consider the local microbiota as a whole [32]. Therefore, this state-of-the-art approach was applied in the present study. Here, a NGS strategy using V3/V4 hypervariable regions, which combines advantages in sequencing depth, error rates, costs, and comparability with other studies, was applied [33]. From the perspective of molecular identification and quantification of defined disease-related taxa (such as *P. gingivalis* or *P. intermedia*), real-time quantitative polymerase chain reaction (qPCR) is the method of choice [34,35]. However, the present study focused on changes in the composition of the microbiota due to the different therapies and not on determining the absolute bacterial cell count. 

The requirement for evaluating complex longitudinal study designs is inclusion of microbiota community variability [36]. A current approach reflecting microbiota diversity within a single sample is to assess alpha diversity over time after therapeutic intervention [36]. Shannon’s index, as one index for detecting alpha diversity, combines richness and diversity, implying that it reflects the variety of species and also takes the inequality between species abundance into account. In the present study, the alpha diversity decreased continuously after Q-SRP and FMDAP interventions (n.s., Figure 2). On the other hand, an increasing trend in alpha diversity after treatment was observed in patients receiving FMS or FMD therapy, although a continuous increase across all study time points was only demonstrated in FMD patients. 

Two opposing hypotheses exist with regard to alpha diversity. One theory is that microbial diversity is higher in the subgingival plaque of patients with periodontitis [37,38], which is attributed to the greater pocket depth affecting the availability of nutrients and/or the impairment of the immune system, allowing the growth of a more diverse microbial community. In line with this theory, Johnston et al. detected a decrease in alpha diversity after nonsurgical periodontal treatment [20]. 

The second theory describes from a bacterial perspective that greater community diversity is associated with increased ecosystem resilience and a healthier condition [32]. This is in line with results of further studies that could confirm a disease-related decrease in alpha diversity compared to periodontally healthy individuals [39,40]. With the understanding that successful nonsurgical periodontal therapy should promote the subgingival microbiota from dysbiosis to a healthier bacterial situation, an increase in alpha diversity should be expected (contrary to the theory introduced by Shi et al., 2018 [37] and Abusleme et al., 2013 [38]). In summary, however, no conclusive assessment of the therapies implemented in this study can be derived due to the different explanatory models regarding alpha diversity.

It should be noted that alpha diversity values are not indicative of changes in microbial community composition [36]. For example, a microbial community can potentially completely shift its composition without sharing similar taxa, but still have similar alpha diversity dimensions. Therefore, in order to assess the clinical success of periodontal therapy in relation to the subgingival microbiota arrangement, changes in community composition (beta diversity) need to be evaluated. In the present study, weighted UniFrac analysis, including abundance and variety of species, was used to assess differences in beta diversity. Here, we could show an increase in beta diversity 3 months after nonsurgical periodontal therapy for all treatment groups (Q-SRP (n.s.), FMS (*p* = 0.034), FMD (*p* = 0.006), and FMDAP (*p* = 0.039), Figure 3). This is consistent with the clinical periodontal improvement measured in the four treatment groups in relation to the proportion of nonbleeding, shallow sites at 3 months (Table 2). In the FMS and FMDAP groups, beta diversity has increased significantly compared to baseline, even 6 months after treatment. Consistent with the increase in beta diversity, these two groups maintained their proportion of nonbleeding, shallow sites during the 6-month follow-up period (Table 2). However, in the FMD group, the rate of nonbleeding, shallow sites had decreased 6 months after treatment, which was accompanied by a significant decrease in beta diversity, too (Figure 3, Table 2). In the Q-SRP group, there was also a very slight but not significant trend for beta diversity to increase over the time, which was associated with an increase of proportion of nonbleeding, shallow sites 3 months after treatment. If previous studies on this aspect are taken as reference, the study designs and data are indeed inconsistent. On the one hand, clinical studies showed that changes in beta diversity were not in line with the periodontal outcome after nonsurgical periodontal therapy [19,41]. It should be noted, however, that none of these studies investigated different nonsurgical treatment approaches over 6 months as in the present study. On the other hand, Feres et al. showed that there was a clear distinction between pre- and post-treatment patients in terms of beta diversity [21]. However, a possible bias might have been introduced by the fact that patients were also treated with antibiotics in addition to nonsurgical periodontal therapy [21]. 

A more detailed assessment of the changes in microbiota composition reveals no significant changes in the Q-SRP group after treatment at all taxonomic levels. This is in line with the pattern of beta diversity, which, as mentioned above, has widely remained unchanged. A different pattern emerged in the other treatment groups. For these treatment groups, a decrease in pathogenic bacteria 3 months after treatment was initially shown, which was associated with a renewed increase 6 months after therapy. In the FMS group, *Pg*, the periodontal key pathogen [42], was significantly affected by periodontitis treatment. It has already been shown that the subgingival occurrence of *Pg* can be beneficially reduced by nonsurgical periodontal therapy [14]. Furthermore, in the FMD group differences in the prevalence of *Eubacterium nodatum* [43] and *Prevotella dentalis* [44] were noted. In the FMDAP group, uncultured *Prevotella* sp. also followed the course of the aforementioned bacteria (decrease 3 months after treatment and then an increase 6 months after therapy). However, it must be noted here that the uncultivated *Prevotella* sp. also include species that do not exhibit periodontal pathogenicity. These results regarding alterations in the microbiota over time after nonsurgical periodontitis treatment, including the gradual reintroduction of the baseline situation, have already been outlined in other studies of periodontitis treatment [45]. Based on the data obtained in the present study, we cannot confirm that the treatments investigated effectively perturbed the dysbiotic composition of the oral microbiota, which remains a condition associated with periodontal health to be promoted in the long term.

### Study Limitations

The overall aim of this study was to determine longitudinal changes in clinical and microbial parameters after various nonsurgical periodontal treatments. However, there are certain limitations to the study design and the results should be interpreted in consideration of these factors. 

One limiting factor is the lack of a healthy control group. It should be of interest not only to determine changes in the post-treatment microbiota but also to compare these changes with the microbiota associated with pristine periodontal health. Further studies should address this issue. 

In addition, the patients were evaluated only at baseline and 3 and 6 months after treatment. By increasing the frequency of the examination intervals, possible shifts in the microbiota might be detected at an earlier stage. This knowledge could contribute to improved staging of a periodontal intervention. In a randomized clinical trial, Socransky et al. followed up patients over a period of 24 months after treatment and demonstrated in checkerboard DNA-DNA hybridization analyses that the change in select subgingival bacteria contributing to the stability of the periodontium was ensured over the study period [46]. Considering the challenge of keeping the periodontium healthy over a long period of time and reliably detecting a recurrence of periodontitis, studies with longer follow-up times should be conducted. Furthermore, no other factors that could influence periodontitis, its progression, and the outcome of nonsurgical periodontal treatment were considered in this study. For this reason, further large-scale studies should include the assessment of known periodontal risk modulators [47] such as smoking [48], gender [49], socioeconomic status [50], or oral hygiene management [51,52]. Therefore, these findings and their relevance for clinical practice should be considered on an individual basis [53].

Another limitation is that other substances whose effects on oral microbiota have already been demonstrated were not included in this study. Substances including paraprobiotics [54]; polyphenols, essential oils, and alkaloids [55]; and tea tree oil and ethanolic extract of propolis [56], aloe vera [57], or ozone [58] can be considered important adjuncts to periodontal therapy. For this reason, further studies should investigate such options for supplementing periodontal therapy.

## 5. Conclusions

In conclusion, we could confirm our hypothesis that the clinical success of nonsurgical periodontal treatment is associated with a change in microbial composition. The proportion of nonbleeding, shallow sites assessed by periodontal treatments with a full-mouth approach (FMS, FMD, and FMDAP) was accompanied by a reduction in the subgingival occurrence of bacteria presumably associated with periodontitis. Despite further improvement (FMS and FMDAP) or a slight decrease (FMD) in clinical outcomes, there was a clear recurrence of bacteria 6 months after therapy.

However, at this time, no clinical implications can be derived from the available microbiological data. The results from this study can be used to design large-scale prospective studies to investigate different nonsurgical periodontal treatment strategies, considering further influencing periodontal factors on the microbiota profile. This may open up opportunities to integrate the subgingival microbiota for diagnosis and risk assessment of periodontitis in the future. Investigating the subgingival microbiota might help appraise successful periodontal treatment and implement individualized therapy.

## Figures and Tables

**Figure 1 jcm-11-01187-f001:**
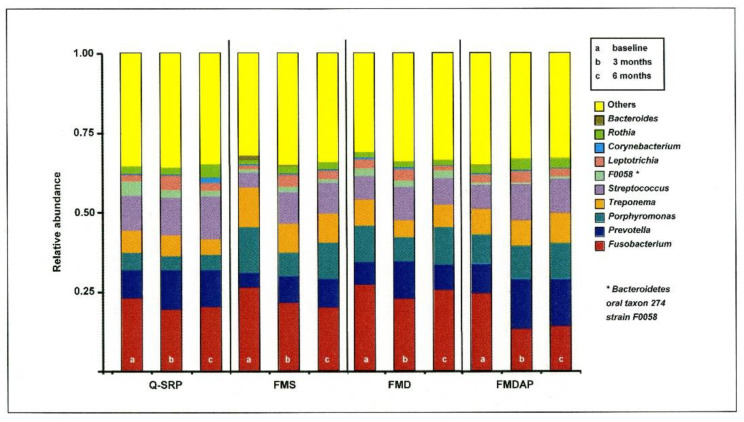
Microbial community composition at genus level (top 10 with the highest relative abundance). The top four genera were *Fusobacterium*, *Prevotella*, *Porphyromonas*, and *Treponema*. No statistically significant differences were identified among these genera within the periodontal treatment groups over time. Q-SRP: quadrant-wise subgingival scaling and root planing (SRP); FMS: full-mouth SRP without the use of antiseptics; FMD: full-mouth SRP within 24 h with disinfection using antiseptic applications; FMDAP: FMD combined with the use of subgingival erythritol air-polishing.

**Figure 2 jcm-11-01187-f002:**
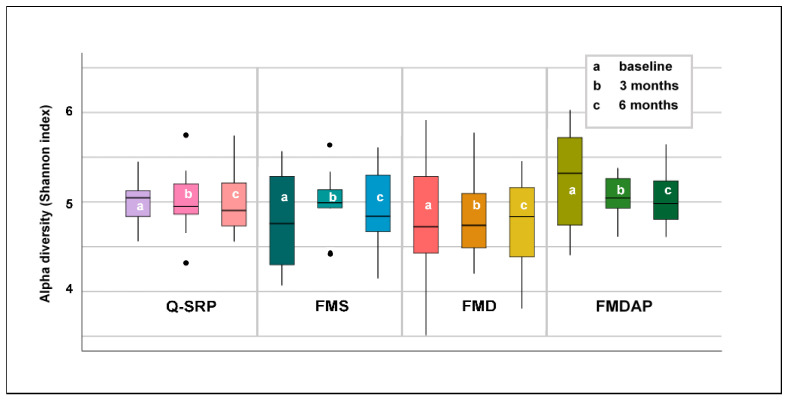
Variations over time in alpha diversity (Shannon index) of patients’ subgingival microbiota considering periodontal treatment (comparing paired groups by applying Wilcoxon signed-ranks test). Q-SRP: quadrant-wise subgingival scaling and root planing (SRP); FMS: full-mouth SRP without the use of antiseptics; FMD: full-mouth SRP within 24 h with disinfection using antiseptic applications; FMDAP: FMD combined with subgingival erythritol air-polishing.

**Figure 3 jcm-11-01187-f003:**
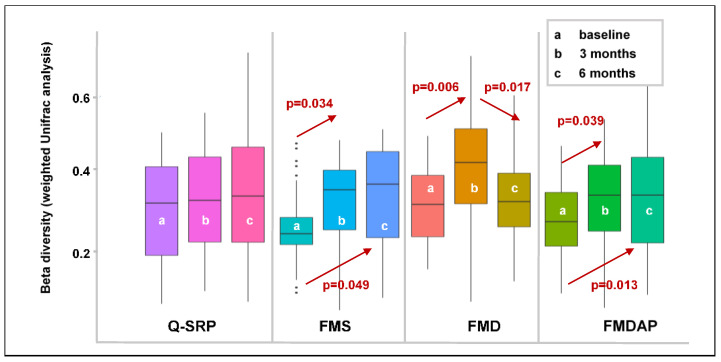
Variations over time in beta diversity (weighted UniFrac analysis) of patients’ subgingival microbiota considering periodontal treatment (comparing of paired groups by applying Wilcoxon signed-ranks test, corrected for multiple comparisons using the Benjamini–Hochberg false discovery rate of 5%). Q-SRP: quadrant-wise subgingival scaling and root planing (SRP); FMS: full-mouth SRP without the use of antiseptics; FMD: full-mouth SRP within 24 h with disinfection using antiseptic applications; FMDAP: FMD combined with the use of subgingival erythritol air-polishing.

**Table 1 jcm-11-01187-t001:** Patient data recorded at the baseline examination.

Variable	Q-SRP (*n* = 10)	FMS (*n* = 10)	FMD (*n* = 10)	FMDAP (*n* = 10)
Age (years)	58.1 ± 9.7	55.3 ± 11.8	58.9 ± 14.3	58.0 ± 12.6
Gender (male/female)	8/2	6/4	4/6	5/5
Smokers (*n*)	2	4	4	1

Q-SRP: quadrant-wise subgingival scaling and root planing (SRP); FMS: full-mouth SRP without the use of antiseptics; FMD: full-mouth SRP within 24 h with disinfection using antiseptic applications; FMDAP: FMD combined with the use of subgingival erythritol air-polishing.

**Table 2 jcm-11-01187-t002:** Full-mouth clinical parameters at baseline and follow-up visits.

Variable	Timepoint	Q-SRP (*n* = 10)	FMS (*n* = 10)	FMD (*n* = 10)	FMDAP (*n* = 10)
PI	Baseline	1.09 ± 0.35	1.35 ± 0.54	0.98 ± 0.60	1.14 ± 0.48
	3 months	0.55 ± 0.35 *	0.78 ± 0.45 *	0.52 ± 0.46 *	0.49 ± 0.37 *
	6 months	0.60 ± 0.48 *	0.92 ± 0.45 *	0.82 ± 0.54	0.40 ± 0.25 *
GI	Baseline	1.28 ± 0.38	1.47 ± 0.68	1.22 ± 0.55	1.31 ± 0.47
	3 months	0.71 ± 0.56 *	0.79 ± 0.58 *	0.55 ± 0.52 *	0.55 ± 0.45 *
	6 months	0.86 ± 0.68 *	0.64 ± 0.51 *	0.56 ± 0.43 *	0.42 ± 0.32 *
PPD (mm)	Baseline	3.77 ± 0.74	3.92 ± 0.64	4.00 ± 0.49	3.91 ± 0.60
	3 months	3.32 ± 0.66	3.43 ± 0.60 *	3.47 ± 0.47 *	3.18 ± 0.69 *
	6 months	3.36 ± 0.62	3.35 ± 0.58 *	3.30 ± 0.55 *	3.18 ± 0.63 *
CAL (mm)	Baseline	4.36 ± 0.96	4.47 ± 0.52	5.10 ± 1.02	4.68 ± 0.75
	3 months	3.98 ± 0.82 *	4.14 ± 0.49 *	4.66 ± 0.93 *	4.12 ± 0.65 *
	6 months	4.01 ± 0.81 *	4.11 ± 0.64	4.63 ± 1.07 *	4.15 ± 0.63 *
BOP (%)	Baseline	33.17 ± 14.82	49.60 ± 23.00	33.78 ± 14.54	42.36 ± 19.26
	3 months	18.06 ± 11.88 *	24.65 ± 20.54 *	18.73 ± 12.12 *	15.19 ± 15.65 *
	6 months	21.07 ± 11.00 *	17.61 ± 17.43	24.90 ± 17.67 *	12.59 ± 13.78 *
NBSS (%)	Baseline	57.00 ± 17.17	40.50 ± 21.11	47.50 ± 16.88	48.40 ± 18.93
	3 months	71.10 ±13.61 *	64.50 ± 17.78 *	70.40 ± 15.31 *	75.50 ± 19.23 *
	6 months	69.80 ± 12.39 *	70.80 ± 17.55 *	66.6 ± 17.27	79.20 ± 16.40 *

* Statistically significant differences (*p* < 0.05, *t*-test, paired sample) in comparison to baseline values; Q-SRP: quadrant-wise subgingival scaling and root planing (SRP); FMS: full-mouth SRP without the use of antiseptics; FMD: full-mouth SRP within 24 h with disinfection using antiseptic applications; FMDAP: FMD combined with the use of subgingival erythritol air-polishing; PI: plaque index; GI: gingival index; PPD: probing depth; CAL: clinical attachment level; BOP: bleeding on probing. NBSS: nonbleeding, shallow sites (PPD ≤ 4 mm without BOP).

**Table 3 jcm-11-01187-t003:** Occurrence of bacteria at baseline and follow-up visits in different treatment groups (only species are displayed that exhibit significant changes in the *t*-test, paired sample).

Treatment	Time Point	Bacterial Species	
		*P. gingivalis* change regarding baseline	*C. consius* change regarding baseline
FMS	Baseline 3 months 6 months	100% 41.7% * 75%	100% 128.6% * 300% *
		*E. nodatum* change regarding baseline	*P. dentalis* change regarding baseline
FMD	Baseline 3 months 6 months	100% 33.3% * 66.7%	100% 25% * 50%
		Uncultured *Prevotella* sp. change regarding baseline	
FMDAP	Baseline 3 months 6 months	100% 29.4% * 76.5%	

* Statistically significant differences (*p* < 0.05, *t*-test, paired sample) in comparison to baseline values. FMS: full-mouth SRP without the use of antiseptics; FMD: full-mouth SRP within 24 h with disinfection using antiseptic applications; FMDAP: FMD combined with the use of subgingival erythritol air-polishing.

## Data Availability

The comprehensive study protocol is available from the authors. Data can be provided by the authors upon request.

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
