# Peer review of "Nonsurgical Periodontal Treatment Options and Their Impact on Subgingival Microbiota"

_jcm, 2022, doi:10.3390/jcm11051187_

Round 1

Reviewer 1 Report

The authors to compare the microbiological outcome of different clinically successfully applied non-surgical periodontal therapies (Q-SRP, FMD, FMS, FMDAP) in patients with 72 periodontitis stage III/IV using Next Generation Sequencing methods.

The study covers some issues that have been overlooked in other similar topics. The structure of the manuscript appears adequate and well divided in the sections. Moreover, the study is easy to follow, but few issues should be improved. Some of the comments that would improve the overall quality of the study are:

1-) The manuscript needs grammar correction. Please also check typos thorough the text;

2-) In the discussion section, the authors stated that: “further large-scale studies should include the assessment of known periodontal risk modulators [46] such as smoking [47], gender [48], socioeconomic status [49], or oral hygiene [50].

I suggest to rephrase as:

For this reason, further large-scale studies should include the assessment of known periodontal risk modulators [46] such as smoking [47], gender [48], socioeconomic status [49], or oral hygiene management [50,51]. Therefore, interpretation of finding and relevance to clinical practice should be considered on individual bases [52].

(please see for [51]: DOI 10.3390/app11167180  ; and for [52] DOI: 10.3390/BIOMEDICINES8050115)

3-) Conclusion Section: This paragraph required a general revision to eliminate redundant sentences and to add some "take-home message".

Author Response

Dear Reviewers,

Thank you very much for your invaluable comments while revising our manuscript. We introduced your comments and critical remarks in the revised manuscript. 

Here, we want to explain how the manuscript was revised. The changes are indicated using blue color.

Reviewer 1

  1. The manuscript needs grammar correction. Please also check typos thorough the text.

The manuscript was proofread for English by native speaker Sherryl Sundell, who was managing editor of the International Journal of Cancer and who has many years of experience as a professional editor.

  1. Discussion:

We would like to thank the reviewer for his/her valuable suggestions for rephrasing, which we have implemented.

  1. Conclusions: This paragraph required a general revision to eliminate redundant sentences and to add some "take-home message".

The conclusions have been carefully revised and condensed. Important implications that are valuable for dental practice have been added.

Reviewer 2 Report

Thank you very much for allowing me to review this manuscript.
some missing parts must be integrated.

Specify in the abstract which bacteria reappear after 6 months

Add specific keywords, these are few.

Sufficient introduction

Materials and methods:
How was the sample size calculated?
What mechanical or manual tools were used for scaling and root planing?
At home, why was an electric toothbrush not recommended? its effectiveness is now amply demonstrated!
What kind of antiseptic was used?
Why did you only focus on the microbiome and not the microbiome?

Correct results

Discussions: to add all the proactive therapies present in the literature that have conducted similar studies with microbiological analyzes, which use natural substances that reduce the bacterial load such as: aloe vera, tea tree, ozone therapy, probiotics or paraprobiotics.
I have a reference from Scribante et all, but there are others in the literature to quote: DOI 10.3390/microorganisms10020337

Conclusions: to reformulate how the bacterial load changes after 6 months

Author Response

Dear Reviewers,

Thank you very much for your invaluable comments while revising our manuscript. We introduced your comments and critical remarks in the revised manuscript. 

Here, we want to explain how the manuscript was revised. The changes are indicated using blue color.

  1. Abstract: Specify in the abstract which bacteria reappear after 6 months.

In the abstract (results) we stated: „In three (treatment) groups, there was a decrease in specific bacteria after 3 months, which was associated with a renewed increase after 6 months (FMS: Porphyromonas gingivalis; FMD: Eubacterium nodatum, Prevotella dentalis; FMDAP: uncultured Prevotella sp.).” All the bacteria mentioned here had decreased in the different treatment groups initially at 3 months after therapy but had increased again subgingivally at 6 months. In the conclusion section of the abstract the bacteria names were not repeated again because the word count of the abstract is limited.

  1. Abstract: Add specific keywords, these are few

The list of keywords has been extended.

periodontitis stage III/IV; nonsurgical periodontal treatment; quadrant-wise debridement; full-mouth scaling; full-mouth disinfection; full-mouth disinfection with adjuvant erythritol air-polishing; subgingival microbiota; Porphyromonas gingivalis; Eubacterium nodatum; Prevotella dentalis

  1. Materials and Methods: How was the sample size calculated?

The present study included 40 patients, 10 in each treatment group. This small-scale study is initially a hypothesis-generating study as no comparable study approach has been published so far. If, as a result of this study, one of the four treatment methods is preferable from a microbiological point of view, in addition to showing clinical benefit, then this result could be confirmed in larger multicenter studies and a clear sample size calculation can be performed.

The German Federal Ministry of Education and Research also supports this approach of hypothesis-generating studies. Until 2030, it will fund science-initiated, exploratory clinical trials that provide initial proof of efficacy for a therapy concept with small numbers of subjects and serve to prepare multicenter clinical trials with large numbers of subjects.

  1. Materials and Methods: What mechanical or manual tools were used for scaling and root planing?

In all treatment groups, subgingival instrumentation was performed after local anesthesia and included the use of ultrasonic scalers (Piezon Master®, EMS, Nyon, Switzerland), followed by instrumentation with Gracey curettes (Hu-Friedy, Frankfurt, Germany). In the FMADP group, additional subgingival air-polishing (Air-Flow®, EMS) using erythritol powder (Air-Flow powder plus®, EMS) was applied for 20 seconds/tooth (5 seconds/surface). Thereby, for all pockets an Air-Flow® (EMS) handpiece was used. In sites with PPD >5 mm, additionally the Perioflow® handpiece with the Perioflow® nozzle (both EMS) was used in order to gain access to the deeper subgingival areas.

  1. Materials and Methods: At home, why was an electric toothbrush not recommended? its effectiveness is now amply demonstrated!

All participants were provided with the same toothbrushes and toothpaste (both: meridol®, CP-GABA, Hamburg, Germany) as well as interdental brushes (elmex®, CP GABA). Thus, a possible influence of individual oral hygiene on the study results should be minimized.

  1. Materials and Methods: What kind of antiseptic was used?

Antiseptics were used only in the FMD and FMDAP groups, as follows:

For FMD, full-mouth SRP within 24 hours, with disinfection using antiseptic applications according to the protocol of Quirynen et al. (1998); brushing the dorsum of the tongue with 1% chlorhexidine gel for 1 minute; rinsing twice with 0.2% chlorhexidine solution for 1 minute; application of 0.2% chlorhexidine spray (all GlaxoSmithKline, Munich, Germany) to the tonsils twice; and subgingival irrigation of all pockets with 1% chlorhexidine gel (3 times within 10 minutes), which was repeated after one week. Patients were instructed to rinse twice a day for 1 minute with 0.2% chlorhexidine solution and to spray the tonsils twice a day with 0.2% chlorhexidine spray over a period of 2 months.

For FMDAP, FMD as described above combined with the use of subgingival erythritol air-polishing.

  1. Materials and Methods: Why did you only focus on the microbiome and not the microbiome?

Unfortunately, I'm not quite sure where the little spelling mistake (microbiome vs. microbiome) is at this point? Possibly microbiome vs. microbiota is meant? We focused in our study on the microbiota referring to the subgingival microorganisms. The term microbiome refers to the entire habitat. This term includes the microbiota as well as the genes that comprise them and the environmental factors that influence them. Our study was designed to investigate differences in the composition of microorganisms due to periodontal treatment. In further studies, a more complex approach could be taken and, for example, the metagenome could be included in the investigations. The manuscript was revised in this regard.

  1. Discussion: Add all the proactive therapies present in the literature that have conducted similar studies with microbiological analyzes, which use natural substances that reduce the bacterial load such as: aloe vera, tea tree, ozone therapy, probiotics or paraprobiotics.

Many thanks to the reviewer for this valuable comment. In the discussion of the revised manuscript, the importance of periodontal therapies with natural substances (paraprobiotics [1]; polyphenols, essential oils, and alkaloids [2]; tea tree oil and ethanolic extract of propolis [3], aloe vera [4], and ozone [5]) that also reduce the bacterial load was addressed. These mentioned substances can be considered important adjunct to periodontal therapy.

  1. Butera, A.; Gallo, S.; Pascadopoli, M.; Maiorani, C.; Milone, A.; Alovisi, M.; Scribante, A. Paraprobiotics in Non-Surgical Periodontal Therapy: Clinical and Microbiological Aspects in a 6-Month Follow-Up Domiciliary Protocol for Oral Hygiene. 2022, 10, 337.
  2. Furquim Dos Santos Cardoso, V., Amaral Roppa, R.H., Antunes, C., Silva Moraes, A.N., Santi, L., Konrath, E.L.Efficacy of medicinal plant extracts as dental and periodontal antibiofilm agents: A systematic review of randomized clinical trials. Ethnopharmacol. 2021, 5;281, 114541.
  3. Wiatrak, K.; Morawiec, T.; Rój, R.; Kownacki, P.; Nitecka-Buchta, A.; Niedzielski, D.; WychowaÅ„ski, P.; Machorowska-Pieniążek, A.; Cholewka, A.; Baldi, D.; Mertas, A. Evaluation of Effectiveness of a Toothpaste Containing Tea Tree Oil and Ethanolic Extract of Propolis on the Improvement of Oral Health in Patients Using Removable Partial Dentures. Molecules. 2021, 26, 4071.
  4. Pradeep, A.R., Agarwal, E., Naik, S.B. Clinical and microbiologic effects of commercially available dentifrice containing aloe vera: a randomized controlled clinical trial. J Periodontol. 2012 Jun;83(6):797-804.
  5. Hayakumo, S., Arakawa, S., Takahashi, M., Kondo, K., Mano, Y., Izumi, Y. Effects of ozone nano-bubble water on periodontopathic bacteria and oral cells - in vitro studies. Technol. Adv. Mater. 2014, 15, 055003.

  1. Conclusion: reformulate how the bacterial load changes after 6 months

In order to explain the course of the occurrence of the bacteria after therapy, a table was created and added to the results section. First, 3 months after therapy, the mentioned bacteria were reduced, and 6 months after therapy, an increase in these bacteria was recorded again.

Round 2

Reviewer 2 Report

The manuscript has been properly revised, following the comments of the first revision